# Transformation of financial institutions grants from the government to inclusive financial institutions in Indonesia

**Valeriana Darwis, Rika R. Rachmawati⬤, Chairul Muslim, Chanifah, Asma Sembiring, Nyak Ilham, Lyli Mufidah⬤\*, Sri H. Suhartini, Rofik S. Basuki, Yanti Rina, Suharyon, Maryam Nurdin, Dahya, Mario Damanik, Dina O. Dewi**

Research Center for Behavioral and Circular Economics, National Research and Innovation Agency (BRIN), Jakarta, Indonesia

\* lyli.mufidah@gmail.com

**Data Availability Statement:** All relevant data are within the manuscript and its Supporting information files.

## Abstract

Low-income communities have limited capital and access to money or loans from formal financial institutions. To solve the problems, the government provides solutions, one of them is by forming a microfinance program, namely Rural Agribusiness Business Development (PUAP). PUAP program is one of the grant activities to farmer group association (Gapoktan) with a total capital assistance of *IDR* 100 million. The problem with the 52,186 Gapoktan units that participated in PUAP activities, only 7,703 units (15%) were transformed into Agribusiness Microfinance Institutions (LKMA). This paper differs from others as it briefly explains the PUAP/MFI's institutional transformation and the factors that affect its sustainability, which is so far still limited discussed. The paper aims to see what transformations Gapoktan becomes an inclusive LKMA and the level of sustainability of the LKMA. The research was conducted in Kendal Regency, Central Java- Indonesia, in 2022 on 5 LKMA. The process of transforming LKMA into an inclusive financial institution is analyzed descriptively. LKMA sustainability levels were analyzed using a multidimensional scaling (MDS) approach with the Rapfish application. So far, MDS with the Rapfish application is still very limited for microfinance analysis. MDS analysis is employed because it is relatively simple and effective for looking at sensitive attributes in improving sustainability and generating leverage attributes that can be used for policy-making. The result study shows that the transformation of PUAP into LKMA is driven by the ability to improve legality, financial governance and diversify the customer's business field. The five LKMAs have a sustainability status of 'sufficient' in running their business, with an index value of more than 50%. The study recommends 1) the Indonesian government could assist LKMA in improving its legality and 2) LKMA's management should get training by experts to improve its financial capability to manage the cost saving.

**Funding:** Assistance was provided by the Indonesian National Research and Innovation Agency (BRIN) to conduct the study, which only monitored the study. The funder had no role in study design, data collection and analysis, the decision to publish, the manuscript's preparation, and the publication fee. None of the authors got a salary from the funder.

**Competing interests:** The authors have declared that no competing interests exist.

## Introduction

The financial performance of smallholder farmers is significantly affected by access to credit for business capital. The ease of agricultural lending methods in rural areas can improve access to credit [1], the financial performance of smallholders [2], and farmers' income [3, 4] due to increased productivity [5]. However, the fact is that farmers and poor rural people in developing countries face many obstacles in accessing formal financial institutions [6, 7]. This lack of access is due to the lack of information about the agricultural sector in formal financial institutions [8], banks' orientation is the most significant profit, and not all banks have a network of branch offices in the countryside [9, 10]. The condition causes farmers to choose to borrow money from informal institutions [11] such as credit associations, private loan sharks, relatives [12], traders of production facilities [13] and credit cooperatives [9]. The problem is that loan interest in informal financial institutions is higher than in formal financial institutions [10]. Efforts are needed to overcome the problem by forming inclusive financial institutions to ease farmers' and community members' accessing and using financial services [14].

Inclusive financial institutions have been implemented in several developing countries but face various obstacles. In Vietnam, for example [15], rural credit cannot increase farmers' incomes, and the positive impact is felt only by wealthy households that receive more significant credit. In Pakistan, farmers' credit cannot increase the gross domestic product of agriculture [16]. Meanwhile, in India, there is a tendency to deviate from the mission of financial institutions with indications of dependence on profits and the growing involvement of significant corporate roles [17].

Based on Government accounting technical standard bulletin number 13 concerning grant accounting, a meaning of grant spending is a government spending in the form of cash/goods/service, that could be given to other governments, international organisation, central/local government, state own company, community group, or community organisation that spesifically has been set the use, not compulsory nor binding, also is not continuely except determined differently in the legislative regulation.

The national team of poverty countermeasures acceleration presented that some government empowerment programs provide grants to community organizations. The grant is expected rolling hence it could use continuously. One of the empowerment programs is the national community empowerment program- called *PNPM Mandiri*. PNPM Mandiri's program aims to create sustainable poverty alleviation and job creation in rural areas.

One activity of the *PNPM Mandiri* program in agricultural sector is Rural Agribusiness Development (*PUAP*). PUAP activities are business capital assistance to each farmer group association (Gapoktan) in one village. Each Gapoktan is expected to form a good LKMA, which can drive sustainable financial performance [2].

The *PUAP* activities were carried out from 2008 to 2015 in 52,186 Gapoktan. In its implementation, it is expected that each Gapoktan will form LKMA. The LKMA will manage the savings and loans of its members by following the agreed regulations. However, until now, not all Gapoktan make up LKMA. According to the Directorate of financing of the Ministry of Agriculture, the number of LKMA has formed until 2022 is only 7,703 unit (15%).

The low formation of LKMA is because: (1) establishing institutions is quite complex, and it takes time [18] (ii) the capacity of the management has not been able to manage the turnover of funds, (iii) some farmers are still late and do not return the loan funds because they are considered *PUAP* as the grants from the government, and (iv) the use of funds for consumptive activities [19–21] (and absence of clear instructions on the continuation of the revolving fund program of the *PUAP* program [22]. On the other hand, LKMA has been formed can provide benefits in (i) the provision of productive business capital for members [23], (ii) increased

insight into savings and loans, (iii) increased interaction between farmers, (iv) obtaining loans with easy procedures and conditions, (v) no longer tied to intermediaries or loan sharks and (vi) Adding farmers' income, productivity and employment opportunities [24–29].

This paper aims to determine the process of transforming PUAP institutions into inclusive financial institutions and the level of sustainability factors of LKMA in Kendal Regency. Kendal was chosen because it is one of the districts participating in PUAP activities. The content of the paper is different from others as others are more focused on the impact of lending on the increase of production, sales, assets and profits; relationships between farmers and banks in some countries regarding credit approval and demographic and economic factors affecting access to credit [24, 30–33]. The paper briefly explains PUAP's activities, research locations, PUAP institutional transformation, especially in legality, institutional governance and credit, and the levels and factors that affect its sustainability.

## PUAP activities

The objectives of PUAP activities include: (i) Reducing poverty and unemployment through the growth and development of agribusiness activities in rural areas fits with regional potential; (ii) improving the ability of agribusiness actors, Gapoktan administrators, extension workers and supervisors of farmer partners; (iii) institutional empowerment of farmers and rural economies for the development of agribusiness activities; (iv) improve the institutional function of the farmer economy into a network or partner of financial institutions in the context of access to capital.

PUAP provided capital assistance of Rp. 100,000,000 to one Gapoktan in one village. Each Gapoktan is expected to form a microfinance institution because it is engaged in agriculture. The name of the institution is the Agribusiness Microfinance Institution (LKMA). The establishment of LKMA is intended to: (1) provide certainty of services and ease of access for farmers to financing facilities; (2) provide simple and fast procedures; (3) the proximity of the service location to the farmer's place of business; and (4) LKMA managers fully understand the character of farmers as customers. Later, LKMA is expected to be one of the sources for farmers or rural communities in borrowing business capital [34, 35].

The main task of LKMA is to manage assistance from the government in loans for LKMA members. The lending process starts with the farmer member filling out the Member Business Plan (MBP) form and submitting it to the farmer group. Farmer groups followed the MBP as a Group Business Plan (GBP). The GBP was then handed over to Gapoktan. After that, Gapoktan will issue a Joint Business Plan (JBP). This JBP is used as a loan legality by Gapoktan.

## Methodology

"Human participants by filling out a questionnaire, approved by Research Center for Behavioral and Circular Economics, National Research and Innovation Agency (BRIN)".

The study used the cluster proportional sampling technique. We employ the technique because this is follow-up research carried out in 2018, 2019 and 2022. In 2018, the topic was the evaluation of the farmer group association, which has formed an LKMA and has received a business license from the Financial Services Authority (FSA). The result was that Kendal district was selected as the location with the highest number of best LKMA, namely 8 LKMA. The 2019 research topic related to giving awards to the best LKMA management to get additional business capital from the Ministry of Finance of the Republic of Indonesia. Of the 8 LKMA surveyed in Kendal Regency, after the selection, only 5 LKMAs were entitled to additional capital from the Ministry of Finance through the Government Investment Center (GIC). The

2022 study aims to determine: (i) What transformation has occurred from forming the LKMA until 2022 and (ii) analyze the sustainability of the LKMA business.

## Time and location study

The research was carried out in 2022. A survey of Kendal Regency was carried out in November 2022. Purposive sampling was employed to determine samples. The respondents were 5 LKMA that had received additional capital from the Ministry of Finance through the Government Investment Center: LKMA Mojo Agung, LKMA Anugerah Tani Makmur, LKMA Sendang Mulyo, Koperasi Tani Tamansari and LKMA Blorok Makmur Sejahtera (Fig 1). We verbally asked for the participant's consent to be interviewed as the study respondents. Our respondents ranged from 26 to 60 years old (adults); therefore, they could decide to accept or reject the permission. This study used primary and secondary data. Primary data were obtained through focus group discussions with LKMA administrators. Meanwhile, Secondary data from the Ministry of Agriculture and the Kendal District Agricultural Service.

## Tools analysis

The 2018 study used qualitative descriptive analysis tools focused on LKMA, which was still operating. The 2019 research focused on the supervision and assistance of LKMA management to meet the capital increase requirements of GIC. Meanwhile, research in 2022 looked at what transformations have been carried out by LKMA and what factors affect the sustainability of LKMA.

The level of sustainability of LKMA is analyzed on a multi-attribute basis from various economic, social, environmental, institutional and technological dimensions. The large number of attributes used as indicators makes the analysis process quite challenging to carry out. The right approach to use is Multidimensional Scaling (MDS). MDS is chosen to make large amounts of data or highly complex multivariate data more accessible to interpret [36]. MDS is quantitative-qualitative descriptive research in which a qualitative approach is used to determine the situation and conditions of the existing location through interviews, while the quantitative approach is used to assess sustainability from various aspects of dimensions and what aspects affect the sustainability [37].

The Multi Dimension Scaling (MDS) method has been widely used to observe the sustainability of a business in Indonesia [37–45]. However, the MDS method is rarely used to measure the sustainability of LKMA. Studies in other countries regarding the LKMA's sustainability mostly use an accounting approach (financial performance) [46–48] and some use an institutional approach [49]. The novelty of this study is to analyze the sustainability of LKMA using MDS with the Rapfish application. This method is relatively simple, but the results shown by sensitive attributes can be used by policymakers to improve the sustainability of MFAs.

The MDS analysis tool in this study used the Rapfish 3.1 (Rapid Appraisal for Fisheries) application. The stages of MDS analysis with the Rapfish application include: i) determining attributes, indicators and scoring, ii) determining "good" and "bad" scores and then applying them to Rapfish, iii) multi-dimensional coordination for each attribute iv) sustainability level analysis, v) attribute leverage analysis and vi) model accuracy and conformity analysis (Value of stress, R-squared and Monte Carlo) [50, 51] a distance matrix technique in MDS based on Euclidean distances with the formula:

$$d_{1,2} = \sqrt{(X_1 - X_2)^2 + (Y_1 - Y_2)^2 + (Z_1 - Z_2)^2 + \ldots} \qquad (1)$$

Note:

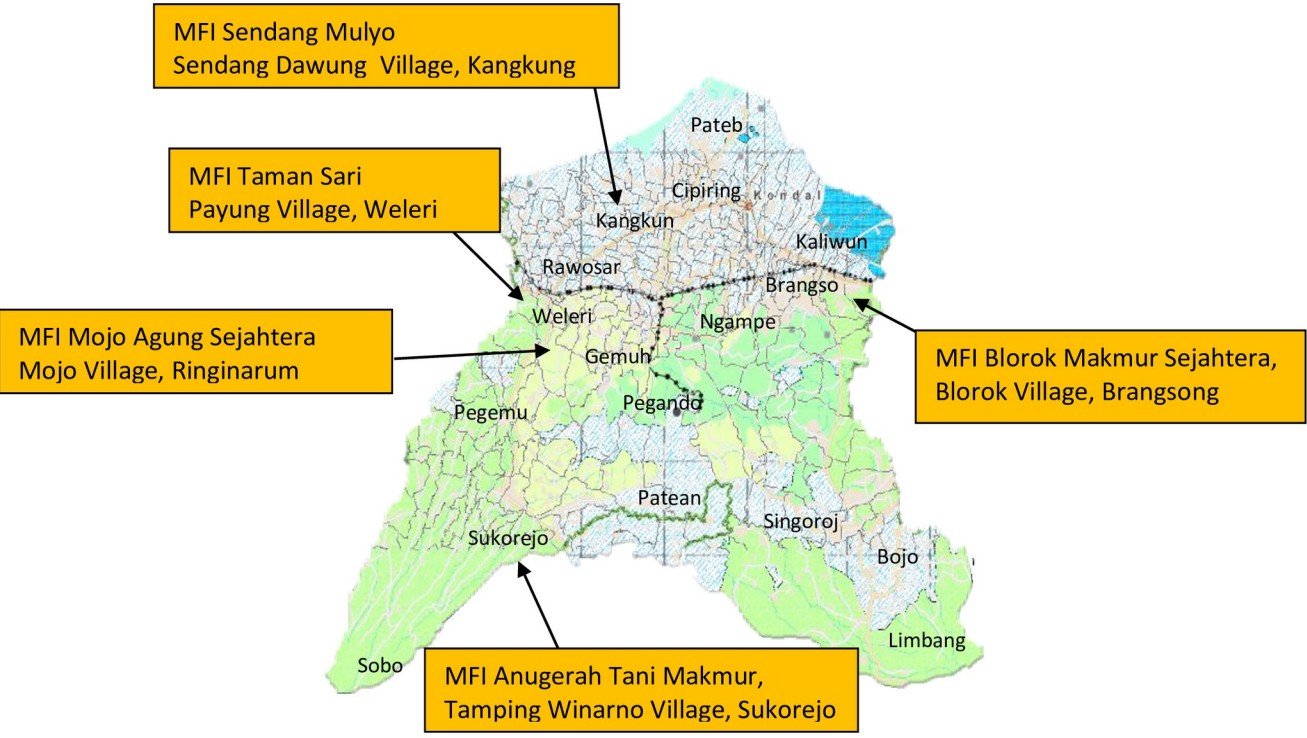

**Fig 1. Research location in Kendal Regency, Central Java province.** Source:https://geoservices.big.go.id/portal/apps/webappviewer/index.html?id= 9917592df1f24501ae804b7d346c08fb (2023). Copyright@2017 Badan Informasi Geospasial All Right Reserved.

$d_{1,2}$ = euclidean distance
X, Y, Z = Atribute
1,2 = observation

The distance matrix is so crucial in MDS [52]; therefore, the distance between two points ($d_{1,2}$) is projected in two-dimensional Euclidean distances with the following regression equation

$$d_{12} = a + bD_{12} + e \qquad (2)$$

Zuhdi [37] states that Rapfish applies the ALSCAL algorithm, which serves as an iteration until it gets a bit error value, as a result of which the intercept value on the equation becomes zero (a = 0) and the equation changes to:

$$d_{12} = bD_{12} + e$$

The iteration process will stop when the voltage value is less than 0.25. Therefore the model is feasible when the stress value is less than 0.25. Voltage values are obtained based on the equation:

$$\text{Stres} = \sqrt{\frac{1}{m}\sum_{k=1}^{m}\left[\frac{\sum_i\sum_j(Dijk - dijk)^2}{\sum_i\sum_j d^2ijk}\right]} \qquad (3)$$

The sustainability of LKMA indicators covered ten dimensions, they are: i) legality, ii) assets, iii) organisation, iv) type of customer business, v) other costs except for interest, vi) LKMA business implementation, vii) number of members development, viii) sources of

income development, ix) expenses, and x) capital and bad debts development. Each attribute's data is converted to ordinal and nominal data through scoring techniques. The scoring technique used a 1–5 Likert scale. Scale 1 represents "bad", and 5 represents "good". The most commonly used distance measures are those calculated from variable pairs using a rating scale such as a Likert-type scale. Table 1 showed the index value of sustainability category.

Attribute leverage determines which attributes are sensitive to sustainability; the higher the attribute value, the more sensitive it is. The most sensitive attributes will contribute to sustainability through Root Mean Square (RMS) changes on the X axis. The Stress and Squared Correlation (RSQ) value is used to determine accuracy in the analysis or Goodness of fit. The Stress value is said to be good if the value is close to 0, while the R-squared is said to be good if the value is close to 1. Monte Carlo analysis is used as a test of the validity and stability of ordinate results. If the ordinate value or difference between Monte Carlo and MDS < 1, then the analysis results can overcome random errors [37, 43, 50, 53].

## Terminology definitions

We used several terminology definitions in our paper, they are Gapoktan, MFIs, LKMA, FSA and GIC.

The farmer groups association (Gapoktan) is defined as a group of several farmers that join and work together to increase economies of scale and business efficiency in one village's administrative area.

Microfinance Institutions (MFIs) are institutions specifically established to provide business development and community empowerment services, either through loans or financing in micro-scale businesses to members and the community, deposit management, and business development consulting services that are not solely for profit.

Agribusiness Microfinance Institution (LKMA) is one of the autonomous business units established and owned by Gapoktan recipients of community direct assistance funds-PUAP in the form of MFIs to solve problems/obstacles to access financial services.

The Financial Services Authority (FSA) is an independent institution free of interference from other parties that organises an integrated regulatory and supervisory system for all activities in the financial services sector.

The Government Investment Center (GIC) is a non-echelon organization responsible to the Minister of Finance through the Director General of Treasury, which disburses Ultra Micro (UMi) loans with an assistance program aimed at UMi business actors who have not been able to access banking financing.

## Results and discussions

### Institutional transformation

The representative of the Central Transformation Office (CTO) of the Ministry of Finance Republic Indonesia, Wempi Saputra, in the presence of the State Wealth Ambassador (KN)

**Table 1. Index and category of sustainability.**

| Index Value | Category |
|---|---|
| 0.00–25.00 | Poor (unsustainable) |
| > 25.01–50.00 | Less (less sustainable) |
| > 50.01–75.00 | Moderate / Enough sustainable |
| > 75.00–100.00 | Good (highly sustainable) |

Source: [37, 52, 53]

(18/5), conveyed that transformation in an institution becomes a necessity because there still many things that can be improved and optimized. Furthermore, in the transformation, two things need to be considered. First, transformation is a process; it cannot be done in a hurry, nor should it be done too late. The transformation must be carefully planned and implemented in earnest. Secondly, transformation is a contribution and will not succeed without the contribution of the smallest unit that wants to transform.

According to Saptana [54], factors that need to be transformed in LKMA include legal entity status, organisational structure, organisational goals or orientation, division of tasks or roles, coordination and communication systems, types of business activities, business management, sources of science and technology, business intensity, human resources skills, and the final product resulting. In line with Saptana's opinion, the institutional transformation seen in the *PUAP* program into an inclusive institution were: changes in legal ownership, changes in organisation and the development of ownership of assets, members and business capital.

There are three stages of the process of transforming Gapoktan into LKMA, first, the transformation of LKMA began when the Gapoktan was formed at the start of the program (2008–2015); second, LKMA under the supervision of the Financial Services Authority (FSA) (2015–2018) and third, LKMA under the supervision of the Government Investment Center (GIC) (2019–2022). As the next step in financial development, the transformation process needs to go hand in hand with socioeconomic development, with an increase in living standards. In developing countries, it can go to the point where they will have formal financial institutions that are "modern" and professional [55].

## Legality

Legality means something that is in accordance with laws and regulations or laws. With a clear legal basis, it is hoped that LKMA actors can be more flexible in developing their businesses, more easily cooperate with other parties such as banks, and attract investors [56]. The transformation of legality types in the five LKMAs surveyed has increased, starting at the time of the program to obtaining capital loans from GIC (Table 2). The types of legality required by LKMA under the supervision of Gapoktan are: (i) Memorandum *of Association/ Articles of Association* (AD/ART), as guidelines or rules in running the organization (ii) Decree of the Village Head, as an acknowledgment of the existence of Gapoktan in the village, (iii) and Decree of the Minister of Agriculture regarding the participation of Gapoktan in PUAP activities.

LKMA is in charge of managing finances and providing loans to the public; therefore, it is coached under the guidance of the FSA. Several legalities that need to be completed by the LKMA management under FSA supervision are a deed establishment issued by a notary, institutional endorsement from the Ministry of Cooperatives, Taxpayer Identification Number and Standard Operating Procedures (SOP) in running their business. Some types of SOPs used include SOPs for lending, SOPs for receiving and closing deposits, SOPs for collection to borrowers, and SOPs for resolving bad debts.

Regulators and supervisors must distinguish between the LKMA of the deposit recipient, whose financial health needs to verify through prudential supervision, and the categories of LKMA that may be registered and subject to non-prudential regulations and standards (e.g. audit reports, consumer information) but do not pose a financial system risk for which the financial authority is responsible [57].

LKMA that gets additional loans from the GIC must complete several other requirements, such as environmental licenses, a cooperative and business registration number, and social health Insurance. Once they are fulfilled, a Letter of Approval for Financing Principles is made

**Table 2. LKMA that has legality (%).**

| No | Types of Legality | LKMA Gapoktan | LKMA FSA | LKMA GIC |
|---|---|---|---|---|
| 1 | Memorandum of Association/ Articles of Association | 100% | 100% | 100% |
| 2 | Decree of the village Head | 100% | 100% | 100% |
| 3 | Decree of Minister of Agriculture | 100% | 100% | 100% |
| 4 | Deed establishment issued by a notary | | 100% | 100% |
| 5 | Endorsement from the Ministry of Cooperatives | | 100% | 100% |
| 6 | Business license from FSA | | 100% | 100% |
| 7 | Taxpayer Identification Number | | 100% | 100% |
| 8 | Standard Operating Procedures | | 100% | 100% |
| 9 | Environmental license | | | 80% |
| 10 | Cooperative registration number | | | 80% |
| 11 | Business Identity number | | | 80% |
| 12 | Social health Insurance | | | 80% |
| 13 | Letter of consent financial principal | | | 80% |

between the manager of LKMA and GIC. In managing all these legality letters, the LKMA management gets assistance from the Directorate Financing of the Ministry of Agriculture, the economic section of the provincial government and the agricultural office district. Not all LKMAs administrators can meet GIC requirements regarding many types of legality needed. One of them, for example, is the administrator of LKMA Sindang Mulyo. As a result, LKMA Sindang Mulyo did not get additional business capital loans from GIC.

## Govarnance

Financial institution governance is a framework in the form of roles, rules and relationships of organs in the institution and their derivatives in directing and controlling businesses to realize goals and objectives. [28] stated that the essential things in the governance of financial institutions are: potency, human resources (management), honesty and transparency in managing capital. Transparency can be represented in the Annual Member Meeting.

Governance in LKMA institutions is more focused on: (i) the governance of LKMA institutions and (ii) the governance of loan implementation. Institutional governance when LKMA was under Gapoktan, the management was not distinguished (Table 3). The reason is that no members are willing to become special administrators of LKMA. Under the supervision of FSA, the management of LKMA is entirely handed over to the manager. It applied to 4 LKMA, or 80% of samples. The secretary and treasurer assist them. The management is growing by adding special information technology (IT) personnel after getting a loan from GIC as it requested LKMA to send borrower requirements and the financial statements report monthly via the internet.

LKMA administrators will be elected at the time of the annual meeting. The length of the management period varies by group. There is one LKMA (20%) for two, three, and five years of management periods found in each of the two LKMAs. Under the supervision of the FSA, LKMA, whose term of office is two years, is only 20 per cent and becomes non-existent when LKMA gets assistance from GIC. When managed by the Gapoktan, LKMA managers get dividend once a year when the Annual Member Meeting is conducted. After that, it became a monthly incentive, especially for 60% of LKMA administrators.

When LKMA is under the GIC, management LKMA gets an honorarium every month becomes 80% Fadikpe [46] suggested that to survive in the industry, LKMA must increase its staff's capacity to handle its customers. The benefits of regulating microfinance may be limited

**Table 3. Overview of LKMA institutional governance (%).**

| No | Jenis Legalitas | LKMA Gapoktan | LKMA FSA | LKMA GIC |
|---|---|---|---|---|
| | **Management** | | | |
| 1 | Farmer group association leader | 100 | 100 | 100 |
| 2 | Farmer group association secretary | 100 | 100 | 100 |
| 3 | Farmer group association treasurer | 100 | 100 | 100 |
| 4 | LKMA manager | | 100 | 100 |
| 5 | LKMA secretary | | 80 | 100 |
| 6 | LKMA treasurer | | 80 | 100 |
| 7 | IT | | | 100 |
| | **Administrator incentives are derived from:** | | | |
| 8 | dividend | 100 | 40 | 20 |
| 9 | Monthly honorarium | | 60 | 80 |
| | **Term of office and election process** | | | |
| 10 | Position period | | | |
| | 2 years | 20 | 20 | 0 |
| | 3 years | 40 | 40 | 40 |
| | 5 years | 40 | 40 | 60 |
| 11 | Selection process through AMM | 100 | 100 | 100 |
| | **Member requirements** | | | |
| 12 | Principal deposits | 100 | 100 | 100 |
| 13 | Mandatory deposits | 100 | 100 | 100 |
| 14 | Domicile within the village | 100 | 100 | 100 |
| 15 | Domicile outside the village in one sub-district | | 60 | 80 |

Source: Primary data, 2022

when commercial banking standards are applied to the LKMA without considering adequate microfinance methodologies. For supervision to be effective, the data requirements and indicators must be adapted to the operations of the LKMA, and they must adapt their information systems to the reporting requirements of the central bank. Given that LKMA's loan portfolio is short-term, stricter standards in certain areas, such as portfolio quality and provision, and more frequent monitoring may be required [58].

Becoming a member of LKMA is not difficult, only paying the principal deposit once and mandatory deposits every month. Apart from being a requirement to become a member, these deposits are also used as additional capital. In general, members' consideration to choose LKMA is influenced by education level, land area, agricultural income, household expenditure, distance from the service provision centre, and perception of interest rates [57]. The difference is about the domicile of members. Previously, it should be from one village. Today, the members' domicile can be outside the village but should be still in the one sub-district.

## Loan governance

The transformation of loan governance changed when LKMA was still joining the Gapoktan management, LKMA under the guidance of FSA and GIC. The change was seen from the borrowers, who used to have to be members of Gapoktan. Wulandari [59] stated that membership is an essential prerequisite to accessing farmer associations' finance. Nowadays, the general community can also make loans for those who live within and outside the village in one sub-district (Table 4). The loan allocation has also changed. Initially, loans were provided for the

**Table 4. Transformation of loan governance of five MFIs in Kendal district (%).**

| No | Types of Legality | LKMA Gapoktan | LKMA FSA | LKMA GIC |
|---|---|---|---|---|
| | **LKMA Debtor Requirements** | | | |
| 1 | Farmer group association member | 100 | 100 | 100 |
| 2 | Public community | | 60 | 80 |
| 3 | Types of loan businesses in the agricultural sector | 100 | 100 | 100 |
| 4 | Types of loan businesses in non agricultural sector | | 80 | 80 |
| 5 | Collateral | | 60 | 80 |
| 6 | The highest loan (000) | 5,000 | 15,000 | 30,000 |
| 7 | Average Interest per month (%) | 1.85 | 1.85 | 2 |
| 8 | Average administration cost | - | 2.2 | 3 |
| | **Development of borrowers and business capital** | | | |
| 9 | Lenders (person) | 778 | 972 | 1.603 |
| 10 | Capital development (000) | 205,800 | 364,200 | 700,200 |

agricultural sector. Nevertheless, loans at currently allocated to agricultural and non-agricultural sector businesses.

Collateral becomes a requirement that is difficult for borrowers to meet farmers mainly borrow funds from LKMA because of the absence of collateral requirements. Collateral is a dominant factor that encourages farmers to access loans [60]. Loosening collateral requirements in lending funds can increase the number of borrowers from farmers [61]. LKMA is the only formal financial institution that can lend money to farmers and is a tool to eradicate poverty [62]. Therefore, collateral was not compulsory when LKMA joined the Gapoktan management. Borrowers only need to make a multilevel application, starting from the Member Business Plan letter given to the farmer group management. Afterwards, the farmer management made a Group Business Plan and submitted it to Gapoktan. The Gapoktan issued a Joint Business Plan containing the amount and timing of the loan and the purpose of using the loan. Under the supervision of the FSA and GIC, the loan application changes with the application for each loan by completing an identity card, filling out a loan contract and being accompanied by collateral such as a land certificate, houses or certificate ownership of motor vehicles.

At the beginning of the PUAP implementation, the highest loan given was only 5,000,000 IDR. The amount was relatively low as LKMA had limited capital [63]. With the capital increasing capital and getting a loan from GIC, LKMA can increase the loan up to 15,000,000, and the latest is 30,000,000 IDR. Initially, the implementation, the loan interest rate averaged 1.85% per month. After that, it rises to 2% per month. Besides interest, borrowers are also charged an administrative fee—initially, 2.2% of the total loan to 3%. In term number of borrowers, there is also a significantly changed. The number of them under the supervision of the FSA increased compared to under the Gapoktan management, from 778 borrowers to 972, and escalated to 1,603 borrowers after becoming GIC customers. The rise in the number of borrowers, loan interest, and administrative costs improved LKMA's capital from an average of IDR 205,800,000 in 2015 (LKMA management was the same as the Gapoktan) to IDR 700,200,000 in 2021 (after obtaining a loan from GIC). The findings show that changes in governance have a positive effect on increasing LKMA's capital.

After LKMA gets a loan from GIC, it has two sources of loan funding, LKMA capital and GIC. GIC loans are known as Ultra Microloans. The steps to submit the GIC loan are as follows: fill in the loan form LKMA has prepared. Then the form is sent to GIC via the internet. GIC will check whether the borrower has a loan from another financial institution. If there is

none, the loan can be disbursed by LKMA after getting GIC approval. The highest loan amount given is IDR 10,000,000, with an interest of 1% per month and an administration fee of 1% of the total loan.

## Sustainability of LKMA

**Sustainability attributes' value.** The assessment of the sustainability level of the five LKMAs was carried out on ten dimensions (Table 5). Each dimension has attributes that can be assessed both qualitatively and quantitatively. The selected tangible and intangible attributes can represent the condition of the LKMA [64], understanding the essential attributes will help invest scarce resources to improve decision-making and increase the return on investment of an institution or company.

The Legality dimension is assessed from 23 supporting attributes, the asset dimension from 7 supporting attributes, the organizational and incentive dimension from 24 attributes, and the business type and loan interest dimension from 16 supporting attributes. Other cost dimensions except interest are assessed from 2 supporting attributes, the business implementation dimension from 7 supporting attributes, and the average percentage development number of members dimension from 3 supporting attributes. The average percentage dimension of income source development is assessed from 6 supporting attributes, the expenditure type dimension from 4 supporting attributes, and the capital development dimension and bad debts from two supporting attributes. The results of the attribute assessment on each dimension are then averaged and applied in Rapfish.

**Index and sustainability status of five LKMAs in Kendal district.** The index and sustainability status of the Five LKMAs are shown in the results of the Rap Analysis (Table 6). The five LKMAs are "moderately sustainable", meaning they are in the middle category (quite good, but their performance must be improved). LKMA sustainability is related to the institution's ability to manage operating costs and operating income [65]. Efforts, strategies and strong support are needed so that the five LKMAs can improve their performance to become "highly sustainable". Investors' interest in sustainable financing is becoming increasingly apparent. In 2020 the number increased by 28% or 20% compared to the previous year (76). Incorporating sustainability in microfinance business activities through strategic management

**Table 5. Dimensions of sustainability level assessment of five LKMAs in Kendal Regency, 2022.**

| Dimension/ Financial Institution Name | 1 | 2 | 3 | 4 | 5 | 6 | 7 | 8 | 9 | 10 |
|---|---|---|---|---|---|---|---|---|---|---|
| | Legality | Asset | Organization and incentives | Type of business and interest on loans | Other fees besides interest | Business activities | percentage the number of members | percentage of source development income | Spending | Capital Development and bad debts |
| 1. Sendang Mulyo | 3.48 | 2.14 | 2.63 | 4.00 | 2.67 | 2.71 | 3.33 | 2.80 | 4.00 | 3.00 |
| 2. Mojo Agung Sejahtera | 4.86 | 3.71 | 3.71 | 3.00 | 3.00 | 4.29 | 4.33 | 3.83 | 1.00 | 4.50 |
| 3. Taman Sari | 4.62 | 3.57 | 3.71 | 4.00 | 5.00 | 4.00 | 1.00 | 2.00 | 3.50 | 1.50 |
| 4. Anugerah Tani Makmur | 4.95 | 4.14 | 4.25 | 4.50 | 4.33 | 4.43 | 4.33 | 3.17 | 3.50 | 3.00 |
| 5. Blorok Makmur Sejahtera | 4.95 | 3.14 | 3.42 | 4.00 | 4.33 | 4.43 | 2.67 | 3.17 | 2.00 | 3.50 |

Source: Primary Data, 2022

Table 6. Index and sustainability status of five LKMAs in Kendal Regency, 2022.

| No | Name of LKMA | Sustainability index | Sustainability Status | Rank |
|---|---|---|---|---|
| 1 | Anugerah Tani Makmur | 72.74 | moderately sustainable | 1 |
| 2 | Mojo Agung Sejahtera | 63.63 | moderately sustainable | 2 |
| 3 | Blorok Makmur Sejahtera | 62.43 | moderately sustainable | 3 |
| 4 | Taman Sari | 60.88 | moderately sustainable | 4 |
| 5 | Sendang Mulyo | 55.20 | moderately sustainable | 5 |

Source: Primary Data, 2022

can generate competitive advantages, have an impact on positive finances, and financial stability to find new sources of financing and service development [66, 67].

Based on the sustainability index, the rank of five LKMAs based on rank 1 (the highest) to rank 5 (the lowest) are as follows; rank 1 was Anugerah Tani Makmur with an index of 72.74, rank 2 by Mojo Agung Sejahtera with an index of 63.63, rank 3 by Blorok Makmur Sejahtera with an index of 62.43, rank 4 by Taman Sari with an index of 60.88, and rank 5 by Sendang Mulyo with an index of 55.20.

The Performance position and sustainability status of the five LKMAs are shown on the rapfish ordination graph (Fig 2). At the centre of the chart is an "o" (round) sign indicating the fifth position of the LKMA. The fifth position of LKMA is in the index range of 50–70. The more significant index value or the object's position getting to the right indicates a higher sustainability status. Meaning that Anugerah Tani Makmur is at the highest level of sustainability, while LKMA Sendang Mulyo is at the lowest level.

Two LKMA positions are close together, namely LKMA Mojo Agung Sejahtera and Blorok Makmur Sejahtera, while the other 3 LKMAs have positions far apart. The proximity of positions between objects indicates that objects have similarities, while positions far apart indicate that objects have differences.

Based on the sustainability index value, LKMA Anugerah Tani Makmur ranks first. LKMA Anugerah Tani Makmur is superior to other LKMAs, especially on the attributes of i) clarity and fairness legality of LKMA cooperatives formally, ii) types of businesses financed are more varied, including food crops, plantation, horticultural, animal husbandry, marketing, and processing of products with a fairly low-interest rate, which is 1.5%, iii) the implementation of the cooperative business, especially the determination of the loan amount is high enough, reach 30,000,000, IDR, the application of collateral, awards, professional sanctions and annual meeting member, iv) a trend of increasing the number of cooperative members from 2013–2021 was positive and kept increasing, v) Professional organizational structure and high position incentives, and vi) The ownership of assets in the form of land while other LKMAs do not have it. Previous research showed that the sustainability of MFI is influenced by the superiority of institutional characteristics and attributes [68].

Legality is one of the factors supporting the sustainability of LKMA [69], as to apply for funding from outside parties such as GIC and banks, the clear legality of LKMA will be questioned. Meanwhile, LKMA finances various types of businesses to avoid the risk of bad debts. If one of the business fields loses, the other business fields are expected to provide benefits, especially if the type of business financing is the agricultural sector. The agricultural business is risky because it is related to external factors that humans cannot control, such as weather, disease attacks and uncertain commodity prices [70, 71].

LKMA Sendang Mulyo is in the last ranking position. The average performance of LKMA Sendang Mulyo attributes below that of 4 other LKMA. Attributes with lower performance in

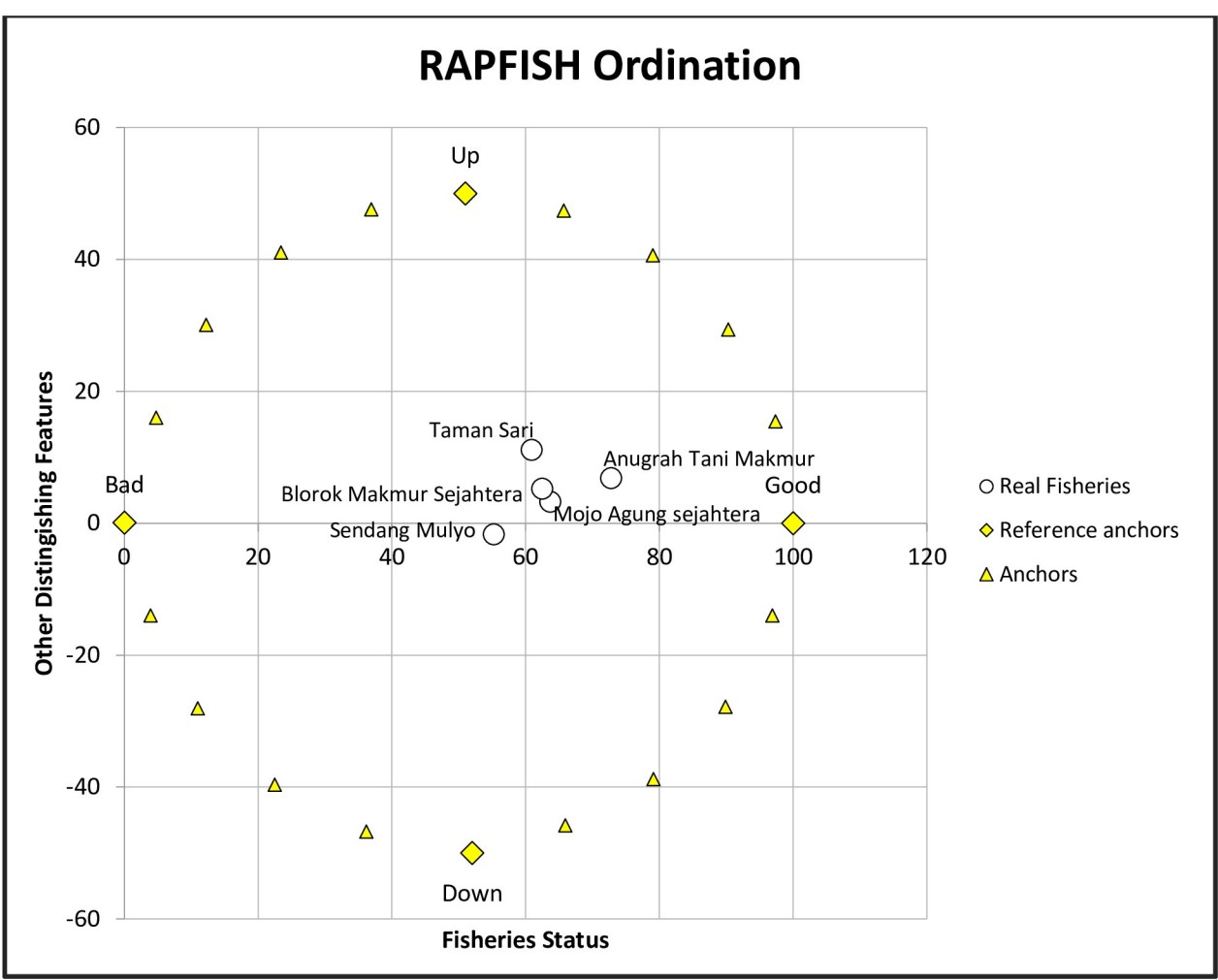

**Fig 2. Rapfish ordination chart at five LKMA institutions in Kendal Regency, 2022.** Source: Primary data, processed 2022. https://ibb.co/GP9tity.

LKMA Sendang Mulyo include i) legality, ii) business implementation, iii) other costs other than interest, iv) organization and incentives, and v) cooperative assets.

**Leverage of attributes analysis.** The attributes that affect the sustainability of LKMA are shown in the Leverage of Attributes Chart (Fig 3). Attributes that most influence the sustainability level of the five LKMAs are: i) Attributes of the average development of the number of members, ii) other costs except an interest, iii) the average development of income, iv) spending, and v) legality.

The number of financial institutions or cooperative members is significant for the institution's progress. Therefore, a development number of members becomes the first attribute that affects the sustainability of an LK MA. Increasing LKMA members will increase mandatory, principal, and voluntary deposits. In addition, it also increases the value of LKMA's accounts receivable, increasing the turnover of capital from loan interest proceeds. [72] Deposit mobilization is a scheme intended to encourage customers to deposit more cash into LKMA so that the money can be used to disburse more loans and generate additional income. Increasing the number of cooperative members must be carried out to improve the sustainability status of LKMA.

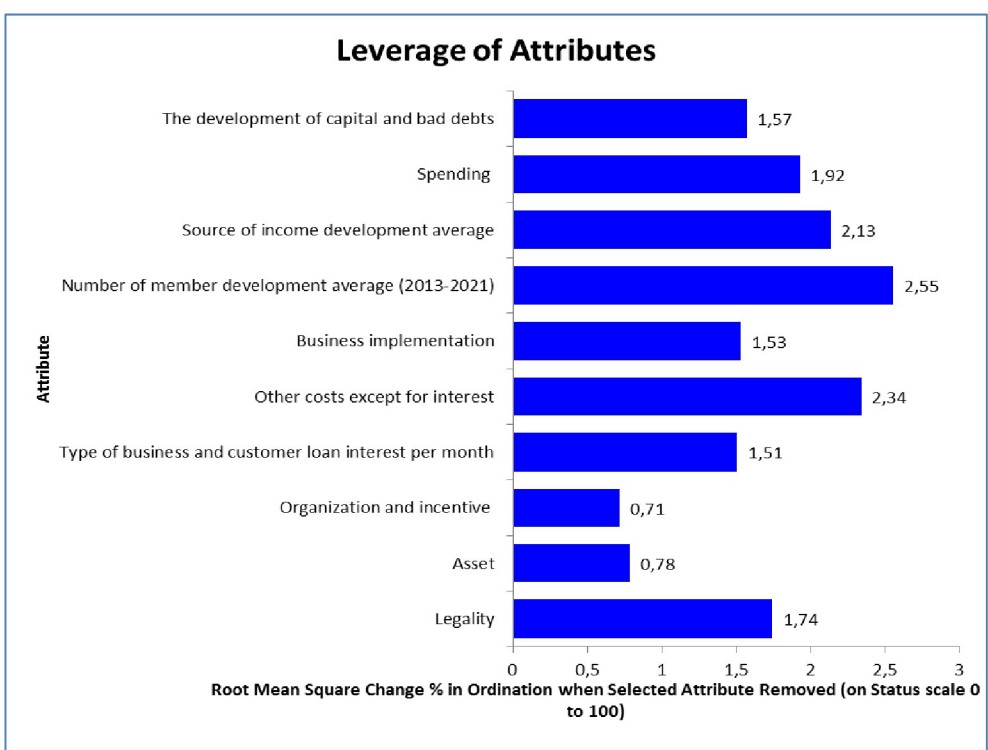

**Fig 3. Leverage of attributes chart on the sustainability level of five LKMA in Kendal District, 2022.**

Other costs, except interest, are the second influences attributed to improving the LKMA sustainability index. The costs consist of i) administrative costs and ii) the presence or absence of cases of late instalment payments. Too high and wasteful administrative costs will affect financial performance and capital availability. The efficiency of administrative costs improves the sustainability status of LKMA because it is more efficient and increases capital. Prominent cases of late instalments from members will reduce capital turnover and LKMA's performance. Improving the sustainability status of LKMA could be done by minimizing instalment delays from members. Members who have income from various sources of business can mostly pay instalments on time 64. [63] stated that the efficiency of administrative costs improves the sustainability status of LKMA because it is more efficient and increases capital.

The third attribute that affects the sustainability of LKMA is the development of sources of income. An increase in the source of income from the addition of mandatory, principal, voluntary deposits and interest on loans will increase capital. LKMA management needs to be creative in increasing income from various sources, such as diversifying loan services to expand cooperative businesses. [73] stated that the provision of loans can increase the dividend, capital, labour and the market affect the income of Small and Medium Enterprise players [72]. The efficiency of LKMA staff members influences LKMA's sustainability in managing borrowers and its ability to use assets to generate income [74]. The higher the source of income of LKMA, the more capital increases. The capital increase will improve the sustainability status of LKMA.

The fourth attribute that affects the sustainability of LKMA is expenditure. [75] microfinance institutions have the challenge of reducing operating costs by minimizing operating costs, declining total expenses, and increasing revenue. Cooperative expenses include

operational costs, management services, office stationary costs, devidend, and annual member meeting costs. The efficiency of cooperative expenses is the primary key to cost savings, especially operational and office stationary costs. Meanwhile, the cost of managing services, dividend, and annual member meeting are hard to be saved because it is already suitable with the number of members.

The fifth attribute that affects the sustainability of LKMA is legality. Legality is the key to increasing members' loyalty, even to increasing the number of members. Formal legality clarity of LKMA will increase the number of members and the level of loyalty. For credit institutions in agriculture, farmers need a solid legal framework and dynamic agricultural development policies to develop microfinance Clarity of legality is one of the factors supporting the sustainability of LKMA [69] to apply for funding from outside parties such as GIC and banks.

## Model accuracy and conformity

The sustainability analysis of five LKMAs in Kendal Regency resulted in a Stress value of 0.177 with a Squared Correlation (RSQ) value of 0.942. A good model is shown with an S-Stress value smaller than 0.25 and an RSQ close to 1. When viewed from the results of Stress and RSQ values, it can be said that the results of the MDS analysis at the sustainability level of five LKMAs in Kendal Regency can represent actual conditions at the LKMA level.

The difference in results between the MDS index and the Montecarlo index reflects the sustainability status. Monte Carlo is used to determining the accuracy of the results. If the difference between MDS and Monte Carlo is <1, then the sustainability index is above 95 per cent. The difference value of <1 indicates that the value of the sustainability index status at the interval confidence according to the RSQ value obtained results that did not have much difference (https://agungbudisantoso.com/multidimensional-scaling-part-4/). The result of the difference between the MDS index and the Monte Carlo index is shown in Table 7.

The five LKMAs showed the difference between MDS and Monte Carlo <1, meaning that they have certainty or sustainability to be developed into a better or more professional LKMA so that their sustainability status increases. The small value of the sustainability index between MDS and Monte Carlo indicates that (1) errors in scoring each attribute are relatively small, (2) the variety of scoring each attribute is relatively small, (3) the repeated analysis process is stable, and (4) errors in the inclusion of missing data can be avoided.

## Conclusion

Five LKMAs that received business capital loans from the Government Investment Center of the Ministry of Finance in Kendal Regency are moderately sustainable and can be continued. It is shown from the MDS index result is more than 50. Meanwhile, attributes that could affect the sustainability of the five microfinances are i) the increasing lender number, ii) the increase in other costs besides interest at the time of borrowing, and iii) the development type of microfinance institutions source of income, (iv) MFI expenditure, and (v) legality. MFI management could maintain and improve sustainability by ameliorating financial performance. It can be done by increasing efficiency through cost savings and the amount of capital supported by strengthening aspects of MFI legality.

The Agribusiness Microcredit Institution, formed through government grants, can slowly transform into an inclusive institution providing business capital loans for the communities' economy. The transformation of the institution through the process of learning and escort in its implementation. The most critical processes include: (i) completing legality requirements as a financial institution, (ii) clear and transparent governance represented from financial

**Table 7. The difference value in the sustainability index of the five LKMAs is based on MDS and Monte Carlo.**

| No | LKMA | Indeks MDS | Indeks Monte carlo | The difference |
|---|---|---|---|---|
| 1 | Anugerah Tani Makmur | 72.73782349 | 72.11051 | 0.63 |
| 2 | Mojo Agung Sejahtera | 63.63219833 | 63.80283 | 0.17 |
| 3 | Blorok Makmur Sejahtera | 62.43326569 | 62.23053 | 0.20 |
| 4 | Taman Sari | 60.87998962 | 60.41442 | 0.47 |
| 5 | Sendang Mulyo | 55.19674301 | 55.4328 | 0.24 |

Source: Primary Data, 2022

statements, (iii) knowing and utilizing information technology in operationalizing work and (iv) escorting and assisting continuously, both from the central and local governments, especially in completing administrative requirements, management training and financial reporting regularly. We recommend that the Indonesian government could assist LKMA in improving its legality and LKMA's management should get training by experts to improve its financial capability to manage the cost saving.

The limitation of this study is that the study was only conducted in Kendal Regency, Central Java, Indonesia, while spatial resource conditions vary between districts. The PUAP program is implemented almost in all districts in Indonesia. Therefore, similar studies are needed in other regions to confirm the results of this study so that it can be applied more generally both in Indonesia and other countries.

## Supporting information

**S1 Data.**
(XLS)

**S2 Data.**
(XLS)

## Author Contributions

**Conceptualization:** Valeriana Darwis, Rika R. Rachmawati, Chairul Muslim, Chanifah, Asma Sembiring, Nyak Ilham, Lyli Mufidah, Sri H. Suhartini, Rofik S. Basuki, Yanti Rina, Suharyon, Maryam Nurdin, Dahya, Mario Damanik, Dina O. Dewi.

**Data curation:** Valeriana Darwis, Rika R. Rachmawati, Chairul Muslim, Chanifah, Asma Sembiring, Nyak Ilham, Lyli Mufidah, Sri H. Suhartini, Rofik S. Basuki, Yanti Rina, Suharyon, Maryam Nurdin, Dahya, Mario Damanik, Dina O. Dewi.

**Formal analysis:** Valeriana Darwis, Rika R. Rachmawati, Chairul Muslim, Chanifah, Asma Sembiring, Nyak Ilham, Lyli Mufidah, Sri H. Suhartini, Rofik S. Basuki, Yanti Rina, Suharyon, Maryam Nurdin, Dahya, Mario Damanik, Dina O. Dewi.

**Methodology:** Valeriana Darwis, Rika R. Rachmawati, Chairul Muslim, Chanifah, Asma Sembiring, Nyak Ilham, Lyli Mufidah, Sri H. Suhartini, Rofik S. Basuki, Yanti Rina, Suharyon, Maryam Nurdin, Dahya, Mario Damanik, Dina O. Dewi.

**Supervision:** Valeriana Darwis, Rika R. Rachmawati, Chairul Muslim, Chanifah, Asma Sembiring, Nyak Ilham, Lyli Mufidah, Sri H. Suhartini, Rofik S. Basuki, Yanti Rina, Suharyon, Maryam Nurdin, Dahya, Mario Damanik, Dina O. Dewi.

**Writing – original draft:** Valeriana Darwis, Rika R. Rachmawati, Chairul Muslim, Chanifah, Asma Sembiring, Nyak Ilham, Lyli Mufidah, Sri H. Suhartini, Rofik S. Basuki, Yanti Rina, Suharyon, Maryam Nurdin, Dahya, Mario Damanik, Dina O. Dewi.

**Writing – review & editing:** Valeriana Darwis, Rika R. Rachmawati, Chairul Muslim, Chanifah, Asma Sembiring, Nyak Ilham, Lyli Mufidah, Sri H. Suhartini, Rofik S. Basuki, Yanti Rina, Suharyon, Maryam Nurdin, Dahya, Mario Damanik, Dina O. Dewi.

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
