## [Decision Letter · Decision Letter 0]

30 Jan 2023

PONE-D-22-35508Transformation of financial institutions grants from the government to inclusive financial institutionsPLOS ONE

Dear Dr. mufidah,

Thank you for submitting your manuscript to PLOS ONE. After careful consideration, we feel that it has merit but does not fully meet PLOS ONE’s publication criteria as it currently stands. Therefore, we invite you to submit a revised version of the manuscript that addresses the points raised during the review process.

ACADEMIC EDITOR:

I have now received the complete reports from the referees about your submitted manuscript. The referees are a seasoned, well-published scholar whose work and opinion I greatly value. You will see from their remarks that they paid very close attention to your paper. Following the report, I re-read and re-assess it with a fair view.  

I appreciate your efforts in writing the manuscript and find the topic exciting and worth pursuing. As you can see from the detailed reports, the reviewers have good evaluations of the quality of your paper. They suggest a revision due to some remaining concerns that need to be addressed. My assessment is similar so I decided to give you an opportunity to revise your manuscript because I trust that you can address all comments adequately. Please take this chance to improve the quality of the paper to satisfy the referee. Please carefully follow the comments to revise your manuscript and resubmit it for re-consideration for publication. Proofreading is strongly advised to ensure that the manuscript is free of errors and well-written.

I will not repeat the reviewers' comments to avoid confusions, hence, would suggest you reading detailed comments in referees' reports. 

We look forward to receiving your revised manuscript.

Kind regards,

Vu Quang Trinh, PhD

Academic Editor

PLOS ONE

Journal Requirements:

"unfunded studies"

5. Please upload a copy of Figures 2 and 3, to which you refer in your text. If the figure is no longer to be included as part of the submission please remove all reference to it within the text.

6. We note that Figure 1 in your submission contain [map/satellite] images which may be copyrighted. All PLOS content is published under the Creative Commons Attribution License (CC BY 4.0), which means that the manuscript, images, and Supporting Information files will be freely available online, and any third party is permitted to access, download, copy, distribute, and use these materials in any way, even commercially, with proper attribution. For these reasons, we cannot publish previously copyrighted maps or satellite images created using proprietary data, such as Google software (Google Maps, Street View, and Earth). For more information, see our copyright guidelines: http://journals.plos.org/plosone/s/licenses-and-copyright.

Additional Editor Comments:

Please pay more attentions to analyses and method comments, but should not ignore others. 

Reviewers' comments:

Reviewer's Responses to Questions

**Comments to the Author**

1. Is the manuscript technically sound, and do the data support the conclusions?

Reviewer #1: Partly

Reviewer #2: Partly

2. Has the statistical analysis been performed appropriately and rigorously? 

Reviewer #1: Yes

Reviewer #2: Yes

3. Have the authors made all data underlying the findings in their manuscript fully available?

Reviewer #1: Yes

Reviewer #2: Yes

4. Is the manuscript presented in an intelligible fashion and written in standard English?

Reviewer #1: Yes

Reviewer #2: No

5. Review Comments to the Author

Reviewer #1: The topic is very interesting. The authors have done a good job for determining the transformation of MFI from government grants to inclusive financial institutions in Indonesia that can help the funding needs of people’s economic businesses in rural areas. This paper provides some important insights for practitioners not only in Indonesia but also in countries which aim to reduce poverty and creating job opportunities in villages. I appreciate the re opportunity to review the proposed article and hope that my considerations will help to improve the work.

1. The abstract section has clearly mentioned the research objectives. However, it is recommended to add the innovation and contributions, such as the differences between other literatures.

2. Please add line number and page number so that pinpointed changes can be recommend.

3. There are some words for introducing MFI in section of results and discussions, I think it may be easier to read if make these sentences as another section after the introduction part.

4. This paper wrote in the methodology section “The study used the cluster proportional sampling technique. We employ the technique because this is follow-up research carried out in 2018, 2019 and 2022”. It is recommended add the relevance and importance of this method to justify the application in this study or explain clearly why the reason for using this technique is the follow-up research carried out in 2018, 2019 and 2022.

5. This paper used the cluster proportional sampling technique and did a good analysis for the data. However, the study aims to determine the transformation of MFI from government grants to inclusive financial institutions in Indonesia that can help the funding needs of people’s economic businesses in rural areas. I think this method can show the transformation but cannot show the relevance with PUAP activities. And this method cannot show the impact on funding needs of people’s economic business in rural areas either. Thus, some other methods, such as OLS method and event study, should also be used in this research.

6. It is recommended to add limitations and future research ideas in the conclusion section. And the implications for Indonesia from the results should also be added.

Reviewer #2: This manuscript investigates what has been made in the context of the transformation of financial institutions grants from the government to the inclusive financial institution and whether the microfinance institution is sustainable. The complete data, the comprehensive presentation, and the systematic storytelling are my first impressions of this manuscript. Using Indonesia as the object of study and the full elaboration of its figurative meanings are also strengths of the manuscript. Together with the findings, the manuscript’s discussion forms a clear marginal contribution to the existing literature. However, I think the research design may be inaccurate. Specifically, the manuscript has the following issues:

First, the manuscript is too close to the form of a research report. I recommended that the study’s intent be supplemented with a complete description of the manuscript’s structure. A more detailed description of the remainder of the manuscript can be added at the end of the Introduction. Section/Subsection that is not directly related to the central strand of research can be placed in the appendices if necessary.

Second, the manuscript takes Indonesia as the default context but does not sufficiently argue in the background statements in the Introduction section whether Indonesia-specific issues are enough general or representative, despite their undoubted importance. In particular, the manuscript’s assessment of findings and policy implications needs to be more widespread; that is to say, although we use Indonesia as a sample, we would like to obtain a more general summarization, both from the perspectives of results and policy. Solving the first issue helps to improve the present issue.

Third, when presenting the research’s background and arguing its importance, it is recommended to add a general picture of such a subject. For example, plot a general state of the country or the world in Figure 1 as a comparison. As far as I can see, after these elements with a broader perspective are added to the research background statement, the importance and significance of the consequential research contents may become more apparent.

6. PLOS authors have the option to publish the peer review history of their article (what does this mean?). If published, this will include your full peer review and any attached files.

Reviewer #1: No

Reviewer #2: No

---

## [Author Response · Author response to Decision Letter 0]

3 Apr 2023

we have corrected the manuscript according to the reviewer's request, to make the manuscript better, hopefully the improvements are as expected.

to reviewer 1:

An explanation of innovations and contributions to research compared to other publications has been included in the abstract section, such as (lines 17-19) and the use of MDS analysis (lines 21-25). Page numbers and line numbers have been added to the paper (see the manuscript). The explanatory paper on MFIs has been covered in the methodology on terminology definitions. We also add some words that quite often we use in the sentence to make it easier for readers (lines 212-231). The relevance and importance of methodologies utilized in line with the research aim have been detailed in depth in the methodology section (lines 150-190; 203-210). Using cluster proportional sampling procedures, qualitative descriptive analysis techniques, and Multidimensional Scaling (MDS) analysis is sufficient in line with the study aims, limits, and research setting. The OLS approach is not yet necessary in this research. The findings concerning the limits and future research suggestions have been provided. Furthermore, the consequences of research findings for Indonesia have been included in conclusion (lines 554-558).

for reviewer 2:

The paper has been revised based on the reviewer's suggestions, and we tried to make it suit the standard for publication in the journal PLOS ONE (see the manuscript). We conducted the study in response to concerns about PUAP's numerous failures as a government initiative. Hundreds of Gapoktans failed to create MFIs for unknown reasons. This study focused on five successful MFIs changing into autonomous, independent financial institutions. The findings of this study can be utilized to inspire government strategies to make PUAP implementation more successful. We also have added cases in other countries that face almost similar obstacles to Indonesia in the 'introduction section' (lines 48-54). In the introduction, we discussed the issues that farmers in Indonesia and other countries face due to the condition of agriculture and the limited cash available to them. This work can be generalized to other countries for future research (see the manuscript) (lines 48-54).

---

## [Decision Letter · Decision Letter 1]

17 May 2023

Transformation of financial institutions grants from the government to inclusive financial institutions in Indonesia

PONE-D-22-35508R1

Dear Dr. mufidah,

We’re pleased to inform you that your manuscript has been judged scientifically suitable for publication and will be formally accepted for publication once it meets all outstanding technical requirements.

Kind regards,

Vu Quang Trinh, PhD

Academic Editor

PLOS ONE

Additional Editor Comments (optional):

Reviewers' comments:

Reviewer's Responses to Questions

**Comments to the Author**

1. If the authors have adequately addressed your comments raised in a previous round of review and you feel that this manuscript is now acceptable for publication, you may indicate that here to bypass the “Comments to the Author” section, enter your conflict of interest statement in the “Confidential to Editor” section, and submit your "Accept" recommendation.

Reviewer #1: (No Response)

Reviewer #2: All comments have been addressed

2. Is the manuscript technically sound, and do the data support the conclusions?

Reviewer #1: (No Response)

Reviewer #2: Yes

3. Has the statistical analysis been performed appropriately and rigorously? 

Reviewer #1: (No Response)

Reviewer #2: Yes

4. Have the authors made all data underlying the findings in their manuscript fully available?

Reviewer #1: (No Response)

Reviewer #2: Yes

5. Is the manuscript presented in an intelligible fashion and written in standard English?

Reviewer #1: (No Response)

Reviewer #2: Yes

6. Review Comments to the Author

Reviewer #1: (No Response)

Reviewer #2: (No Response)

7. PLOS authors have the option to publish the peer review history of their article (what does this mean?). If published, this will include your full peer review and any attached files.

Reviewer #1: No

Reviewer #2: **Yes: **Fu-Wei Huang

---

## [Editor Report · Acceptance letter]

13 Jun 2023

PONE-D-22-35508R1 

Transformation of Financial Institutions Grants from the Government to Inclusive Financial Institutions in Indonesia 

Dear Dr. Mufidah:

I'm pleased to inform you that your manuscript has been deemed suitable for publication in PLOS ONE. Congratulations! Your manuscript is now with our production department. 

Kind regards, 

on behalf of

Dr. Vu Quang Trinh 

Academic Editor

PLOS ONE